# Spatial and social determinants of the 1857 yellow fever epidemic in Lisbon

Isaac H. Bates[1], Sabrina L. Li [1,2]*, Kris V. Parag[3], Katy A.M. Gaythorpe[3], Ana B. Abecasis[4], Matthew Smallman-Raynor[1‡], Nuno R. Faria[3,5‡]

1 School of Geography, University of Nottingham, Nottingham, United Kingdom, 2 Department of Geography, King's College London, London, United Kingdom, 3 Medical Research Council Centre for Global Infectious Disease Analysis, Imperial College London, London, United Kingdom, 4 Global Health and Tropical Medicine (GHTM), Associate Laboratory in Translation and Innovation Towards Global Health (LA-REAL), Institute of Hygiene and Tropical Medicine, NOVA University of Lisbon (IHMT/UNL), Lisbon, Portugal, 5 Institute of Tropical Medicine, Faculty of Medicine, University of São Paulo, São Paulo, Brazil

☯ These authors contributed equally.
‡ These authors contributed equally.
* lisabrinaly@gmail.com

## Abstract

Despite the availability of a highly effective vaccine, yellow fever virus (YFV) is still endemic in 47 countries globally. Although disease due to YFV was first recorded in 1635, factors contributing to its spread remain poorly understood today. Using archival data from the nineteenth century, we digitalised and mapped the 1857 yellow fever (YF) epidemic in Lisbon, Portugal, to understand how transmission dynamics and spatial and environmental characteristics led to disparities in health outcomes between sociodemographic groups. We modelled the basic and effective reproduction number ($R_0$ and $R_t$) and found that transmission dynamics throughout this pre-vaccination era epidemic are consistent with prevailing estimates ($R_0 \simeq 5$). Transmission peaked at the end of October 1857 when YF was declared an epidemic, then declined until January 1858. YFV killed 4.2% of the population with infection attack rates ranging between 10.3-13.5%. Out of the 34 parishes in urban Lisbon, our hotspot analysis identified 15 statistically significant high-risk parishes near the coastline. Our maps, combined with a digital terrain model, show that the highest number of deaths occurred within connected streets confined in low-elevation built-up areas with homes. We discuss the potential role of wind and temperature in aiding mosquito dispersal across Lisbon, which were believed as the main historical environmental drivers of YF. More people died at home than in hospitals, and although working-aged men accounted for most fatalities, the highest probability of death was found among women working at home. Our study highlights the role of human-environment interactions in shaping a historical YF epidemic in a pre-vaccination urban setting and enhances our understanding of modern-day transmission dynamics.

**Data availability statement:** All data used in this study are accessible via the following link: https://github.com/sabrinalyli/1857LisbonYFV.

**Funding:** SLL acknowledges funding from the British Academy ODA Challenge Oriented Research Grant (IOCRG\100955). KVP acknowledges support (MR/X020258/1) from the MRC Centre for Global Infectious Disease Analysis funded by the UK Medical Research Council. KAMG received funding from Gavi (226727_Z_22_Z) and the Bill & Melinda Gates Foundation (INV 034281 and INV 009125 / OPP1157270). ABA acknowledges funding from Fundação para a Ciência e Tecnologia (UID/04413/2023 and LA-REAL—LA/P/0117/2020). NRF acknowledges funding from the Wellcome Trust (Digital Technology Development Award in Climate Sensitive Infectious Disease Modelling, 226075/Z/22/Z), the Wellcome Trust Dengue and Zika Immunology and Genomics Multi Country Network (DeZi Network, 316633/Z/24/Z), and the MRC–São Paulo Research Foundation partnership award (MR/S0195/1 and FAPESP 18/14389–0). The authors acknowledge funding from the MRC Centre for Global Infectious Disease Analysis (MR/X020258/1), funded by the UK Medical Research Council (MRC). The funders had no role in study design, data collection and analysis, decision to publish, or preparation of the manuscript.

**Competing interests:** The authors have declared that no competing interests exist.

## Author summary

Yellow fever is a mosquito-borne disease transmitted to humans primarily through *Aedes aegypti* in urban areas. Currently, yellow fever is endemic in the tropical regions of South America and Africa. Historically yellow fever caused outbreaks and epidemics in North America and Europe too. Climate change and global connectivity are expanding the potential range of *Aedes aegypti,* increasing the risk of yellow fever re-emergence in Europe. To support future preparedness efforts, we examined a historical yellow fever epidemic that occurred in Lisbon, Portugal, that caused 18,000 cases and 5,652 deaths, and assessed its transmission characteristics using the basic and effective reproduction number. We digitalised and mapped archival data at the neighbourhood level to assess the contributions of both social (population, occupations, and gender dynamics) and environmental (temperature, precipitation, wind, and elevation) factors of the yellow fever epidemic in Lisbon. Research into a European epidemic that occurred 80 years before an effective yellow fever vaccine is useful for understanding population-environment dynamics in a pre-vaccination setting where yellow fever is not established. Our findings could strengthen current arbovirus control efforts and support epidemic preparedness in urban areas.

## Introduction

Yellow fever virus (YFV) is an arbovirus presently endemic to 47 countries across the tropical regions of South America and Africa. The disease's extent is limited by social dimensions and the environmental and physical constraints of its primary mosquito vector, *Aedes (Stegomyia) aegypti*, which predominantly circulate in urban areas, but can also, and are increasingly, invading rural areas [1], perpetuating yellow fever (YF) via the intermediate and urban transmission cycles [2]. Globally, 7.2 billion people are at-risk from [3] four arboviruses (YFV, dengue, chikungunya, and Zika [4]), of which YFV is estimated to be responsible for 109,000–200,000 cases and 30,000–51,000 deaths per year [5,6]. While YFV remains enzootic in Africa and South America [7] primarily through its sylvatic (jungle) cycles [8], resurgences are being observed due to climate change [9,10] and increased frequency in human travel in cities [11,12]. Burden estimates are likely to be conservative as many records of YF are often underestimated [13] due to challenges with detection. Most early cases are asymptomatic and those that do show symptoms can be confused with other diseases such as severe malaria and dengue [14].

A prototype member of the Orthoflavivirus genus and Flaviviridae family [15], YFV may have originated between 1,500 and 3,000 years ago in Central Africa [16,17], before diverging into four genotypes, West African, East African, and South American genotypes I and II [18]. The first recorded epidemics of YF dated back to 1635 from Guadeloupe in the southern Caribbean Sea, after being introduced to the Americas with the slave trade – the mosquitoes having harboured in artificial water storage

onboard ships [19,20,21]. Genetic analysis indicates that the two South American genotypes of YFV diverged from the African genotypes in the seventeenth century [16].

The historical context of YFV is crucial for understanding the global trajectory of YFV. The eighteenth and nineteenth centuries saw repeated YF epidemics in American and European ports – examples including, but not limited to, Lisbon, 1723 [22]; Philadelphia, 1793, killing 9% of the population [23]; Leghorn, Italy, 1804 [24]; Cadiz, 1800, 1813 and 1819 [25]; Rio de Janeiro, 1849 [26]; Barcelona, 1821 and 1870 [27,2]; and outbreaks in Porto in 1850 (3 dead), 1851 (40 dead), and 1856 (63 dead) and Lisbon 1856 (87 dead) [22]. Given that the 1856 Lisbon epidemic had little impact on the city, compared to that in 1857, it is highly unlikely that any immunity occurring from infection would have had a significant impact on the outcomes of 1857. European YF historical epidemics occurred most predominately in Lisbon, which suffered the first European epidemic of YF in 1723, predating the commonly cited European YF origins in Cadiz, 1730 [28]. Although a recurring burden to Lisbon during the eighteenth and nineteenth centuries, the scale of the 1857 YF epidemic, and the subsequent breadth of data reported on such magnitude by the Health Council warrants an in-depth analysis of the epidemic, particularly in light of the potential return of YF and arboviruses to Europe.

Despite being one of the most severe health crises of the nineteenth century [29], the 1857 YF epidemic that took place in Lisbon, Portugal, has received relatively little scholarly attention. During this time, Portugal maintained close connections with Brazil, its former colony, which played a central role in driving the transatlantic slave trade [30]. YF reached South America in the seventeenth century, with one of the first Brazilian outbreaks occurring in Recife in 1685, imported on a cargo ship from the island of São Tomé (in the Gulf of Guinea, west of Africa) [31]. Repeated epidemics in Brazil at this time led to the first health campaign in the 'New World' being implemented in 1691; this included the cleaning of houses and streets through bonfires and fumigation, burials of disease victims distant from populated places, establishment of 'port health police' who would inspect ships and crews, and confinement of slaves and some women [31]. Despite this, YF still became established in Brazil. Franco [31] describes one ship that had five YF victims on the journey between Recife and Portugal in 1691, indicating the long relationship between Portugal, Brazil, and YF. Portugal-Brazil migration and trade intensified in the 1840s [13], contributing to the ignition of YF outbreaks in Lisbon by mobilising vectors from endemic regions of South America [32]. The importation of YF from Brazil to Portugal was formally recognised by the King of Portugal for causing the 1856 Porto outbreak; Brazil also drove YF to Montevidéo, Uruguay, and Buenos Aires, Argentina in 1857 [31], illustrating the reach of YF from Brazil. The advent of steam power in the 1830s halved transatlantic travel time to 15 days [26], which fell within the three-to-five-week lifespan of *Aedes (Ae.) aegypti*, meaning that the vectors could survive the transatlantic journey by breeding within water-containing objects onboard (Maricopa County Environmental [33]). *Ae.aegypti* was the most common vector in facilitating historical YF outbreaks [2,34], and can thrive in warm, humid tropical regions [25]. In recent years, *Ae. aegypti* have reestablished themselves on the Portuguese island of Madeira [35], and another vector capable of transmitting YFV, *Ae.albopictus*, was first detected in Portugal in 2017, and then in Lisbon, 2023, indicating the return of YFV-capable mosquitoes to Portugal [36].

The 1857 YF epidemic in Lisbon predates modern vector epidemiological knowledge and YFV vaccination, which was not introduced until the late 1930s. The transmission of YFV by *Ae. aegypti* was proven in 1881 by Carlos Finlay and the hypothesis was confirmed in 1900 by the Walter Reed Commission [20]. Without the knowledge of the role of the mosquito in disease transmission, those who reported the epidemic played into neo-Hippocratic mid-nineteenth century ideas of epidemiology and medicine, that the variation of meteorological conditions caused illness [25,37] which is reflected in historical data collection and reporting.

Here, we investigate the epidemiology of the 1857 YF epidemic in Lisbon, Portugal, using archival historical data. We characterise transmission by estimating the reproduction number of YFV over the course of the epidemic. We study how the epidemic impacted various sociodemographic groups, mapping how human-environment interactions may have contributed to the spread. Our study contributes to the growing literature focusing on the impact of human-environment interactions on disease outcomes across spatial dimensions. Our analysis using nearly complete data of a relatively unstudied

urban epidemic of great magnitude in 1857, in a region that is non-endemic to *Ae. aegypti*, and consequently YF, provides a strong opportunity to assess the drivers of YF. The 1857 Lisbon epidemic is a seminal example illustrating the relationship that YF has previously had with Europe, and may also have in the future as climate change intensifies. Therefore, it is important to understand how YF may spread in a European setting to support the development of epidemic preparedness strategies. These findings using modern spatial epidemiological methods can help inform our understanding of past and present YF infections amid present-day climate and public health challenges.

## Data and methods

### Study area

Known as 'the city of the seven hills', Lisbon is the capital of Portugal. It sits on the northern bank of the river Tagus and is 30 km east of the Atlantic Ocean [38]. Lisbon's Mediterranean climate varies from dry, hot summers to cold seasons frequented by rain. Although official demographic records in Portugal did not begin until 1864 [39], Lisbon was among Europe's 20 largest cities at the end of the eighteenth century. While 'Lisbon' can be defined as the broader region consisting of neighbouring suburbs, we focus only on the urban, 'city centre'. Home to a population of over 133,000, mid-nineteenth century urban Lisbon comprised of four neighbourhoods (S1A Fig). The population was divided relatively evenly between them: Alfama (27.9%), Bairro Alto (26.5%), Alcantara (25.0%), and Rocio (20.5%). The four neighbourhoods consisted of 34 parishes, which were the lowest administrative units created by a variety of royal and governmental edicts (S1B Fig). Details on the names of each parish are outlined in the S1 Table. The most populous parish was Santa Isabel (3) with 8,764 residents, and the least populous was Santa Cruz do Castello (27) with 583 residents. S1C Fig positions the epidemic within the historical timeline of Lisbon's municipal organisation, helping to contextualise changes in the city's administrative structure over time [40].

### Case data

We extracted data from the 1859 report by the 'Extraordinary Public Health Council of the Kingdom [of Portugal]' (hereafter 'Health Council'), published on 29th September 1857 [22], which can be found at the library of the Institute of Hygiene and Tropical Medicine, University of Lisbon, Portugal. The Health Council's report narrates the epidemic's progress, detailing records and maps on the mode of transmission, treatments, and its origin and termination [22]. This report uses mortality data collected by Marino Miguel Franzini, independent to the Health Council, between 1835–57 and meteorological data since 1815 [38], enabling the Health Council to make comparisons with prior years. Jacob Pretorius had collected the earliest meteorological records of Lisbon from 1777-93 [41], which are also used for reference by the Health Council.

The Portuguese Health Council report was translated to English using Google Translate and manually validated. The main body of the report narrates the environmental and socioeconomic conditions of the YF epidemic in 1857. This is supplemented by 36 tables and one page of charts, along with additional information on the occurrence of YF in Lisbon's garrison, treatments used, perspectives on the origins and causes of the epidemic and the mode of YFV transmission. Of the tabular information included in the report, all relevant data for analysing environmental and sociodemographic drivers of YF were extracted and digitalised.

### Environmental data

Lyons [42] details environmental data collected by John Martin, and the Royal Observatory [43,38], which was used to compare to the data extracted from the Health Council and supplemented our data on historic climatic conditions. Temperature, precipitation, wind, and information on elevation were extracted. We note here that data on wind was limited for the period of the epidemic, with the Health Council making only occasional reference to wind direction.

## Socioeconomic data

To extract Lisbon's population data in 1857 by parish, sex, and age group, we digitalised data from a review by Silveira [44] of privately conducted censuses in Portugal 1801 and 1849. These data were used on the assumption that, although populations had changed between 1849 and 1857, they had not changed greatly. The age ranges detailed in the census differ from those used by the Health Council to describe fatalities. Therefore, we conducted statistical extrapolation on the census data available to deduce populations for the same age ranges that were used by the Health Council, to calculate fatality proportions per age grouping.

We retrieved YF case data by sex, access to healthcare, occupation, and the location of hospitals from the Health Council report. Occupations in nineteenth century Lisbon were categorised into nine occupation groups: Agriculture, Commercial, Industrial, Military, Domestic, Professional, Maritime, Lowest Class, and No Designation. Example occupations for each group are summarised (S2 Table). We calculated case-fatality ratios (CFR) where possible from the Health Council's 'official' records of fatalities, split by sex and access to healthcare (as per records of those treated at home versus hospital).

## Estimating the reproduction number (Rt)

To characterise the transmission dynamics of YFV during the epidemic, we estimated the effective reproduction number ($R_t$), and from it approximated the basic reproduction number ($R_0$). This was done to characterise the epidemic using the Bayesian smoothing method known as EpiFilter detailed elsewhere [45,46]. This approach provides minimum mean squared estimates of time-varying reproduction numbers under a widely used renewal model framework [47]. $R_0$ quantifies a pathogen's epidemic potential by estimating how many secondary cases a single infectious individual would generate in a completely susceptible population. An $R_0 > 1$ indicates epidemic growth, $R_0 \simeq 1$ suggest endemicity, and $R_0 < 1$ signifies a declining epidemic with fewer new infections. $R_t$ generalises $R_0$ to provide the transmissibility across time, while considering immunity and control measures. $R_t$ is computable from the generation time distribution of the disease and the incidence of new cases. We use the initial values of $R_t$ as a proxy for $R_0$. To calculate these measures of transmissibility, we inferred the generation time distribution of YFV using a method detailed in Kraemer et al. [48]. Here we seeded a Poisson-distributed number of infections (mean = two cases) for about two weeks before the first observed cases to account for unobserved early transmission. To characterize the incubation period between infection and the time of symptoms onset, we sampled from a truncated exponential distribution. We determined mosquito infectiousness based on the average lifespan of the *Ae. aegypti* mosquito [48] and the extrinsic incubation period, the time from when a mosquito is infected until it becomes infectious. We modelled the extrinsic incubation period using a Weibull distribution with a shape parameter from Johansson et al., [49] and a mean of 6.9 days [50]. To ensure mosquitoes survive long enough to transmit infection, we retain only incubation periods shorter than their lifespan.

## Digitalising historical administrative boundaries

ArcGIS Pro (Vers. 3.1.0) was used for all digitalisation and mapping. Since no digital map exists for 1857 Lisbon, we digitalised a historic base map of Lisbon drawn by Clarke [51], which illustrates a strong resemblance to the contemporary layout of urban Lisbon, as is described in the Health Council report. The map was imported into ArcGIS Pro and was aligned with the Lisbon actual, using buildings as a reference point via the 'Georeference' tools.

In the absence of accessible shapefiles of Lisbon's nineteenth century parishes, we manually digitalised Lisbon's historical parishes into a shapefile by manually tracing an overlain online image of parishes from 1855 (S2 Fig) retrieved from ATLAS Cartografía Histórica [52]. Then, we manually digitalised a shapefile of Lisbon's neighbourhoods by tracing over the parishes to determine neighbourhood boundaries.

## Mapping analysis

We created choropleth maps using the digitalised parishes to visualise the progression of the disease, indicating the mortality rate per 1,000 persons in each parish. We conducted a Getis-Ord Gi* hotspot analysis, a spatial statistics technique, using Fixed Distance Band Interpolation in ArcGIS Pro (Vers 3.1.0) to identify the distribution of significant hotspots of clustered fatalities between parishes. The use of fixed distance band as a conceptualisation of spatial relationships ensures that all parishes are assessed in the context of at least one neighbouring parish [53]. Assessing each parish in the context of at least one neighbouring parish was done under the assumption that neighbouring parishes would have similar environmental and sociodemographic conditions and thus be expected to produce similar disease outcomes.

To map fatalities reported by the Health Council at the street level, we georeferenced each street-level fatality count to the centroids of 101 streets in urban Lisbon. Each count was represented by a colour corresponding to a quantifiable class of fatalities in intervals of 10. Because our base map was independent of the report and omitted some street names, we validated street locations using Google Maps.

We also mapped the locations of all hospitals in Lisbon, thereby permitting an investigation as to whether this social factor had an impact on the spatial progression of the epidemic.

## Buffer analysis

The Health Council report argues that wind was the main driver of YF during the 1857 epidemic. To investigate the contributions of wind to vector dispersal, we created multiple-ring buffer zones around key locations of transmission such as hospitals, the coastline at customs, and street spot points at 106 m and 533 m. These distances were chosen to represent the average and a 'worst-case' scenario for vector dispersal. 106 m [54] was the mean distance produced from a meta-analysis of 27 studies, acting as a consensus and being deemed appropriate. Other studies of *Ae. aegypti* wind dispersal have identified a range between 154 m and 533 m [55,56,57,58], hence, the highest value was taken as a worst-case scenario. Without the influence of wind, indoor-dwelling *Ae. aegypti* are naturally tethered to within 30–60 m of their hatch site, clustering around homes [59], hence an assumption was made that the wind enhanced transmissivity.

Hospitals were incorporated as a key location in the context of the epidemic. A buffer from the coastline was used to consider the impact of wind on transporting vectors from the source of importation. Street fatality count spot points were also analysed to investigate if those with the highest fatalities were near one another, to suggest that vectors had been blown by the wind to infect nearby streets.

Alongside wind analyses, the impact of elevation was investigated by extracting a Digital Terrain Model (DTM), FABDEM V1-2, from Neal and Hawker [60] at a 30 m resolution. The TIF file used to cover the area of Lisbon was N38W009_FABDEM_V1-2. Our study utilised the DTM to generate a contour map to further examine the relationship between elevation and fatalities. To determine the DTM's representativity of mid-nineteenth century topography, we verified locations identified as being either low or high elevation by the Health Council in [22] and checked the relative elevation against the DTM. For example, the Health Council report on the streets of Rua dos Douradores and Travessa da Palha as being of low elevation (less than 30 m altitude), which is clearly demonstrated in the DTM and contour map. Relationships could be examined between elevation and fatality distribution, whilst considering the impact of elevation on wind.

## Results

### Timeline of the 1857 YF epidemic in Lisbon

We identified records from the Health Council indicating that ship-borne importation likely initiated the dispersal of *Ae. aegypti* capable of transmitting YFV during the 1857 epidemic (Fig 1A). The *Genova* arrived in Lisbon in July 1857, after having quarantined in Belém, Brazil. It was reported to be harbouring passengers in ill-health when they arrived in Lisbon. This occurred in the same month as the first YF cases were reported [22], one being a customs labourer who became

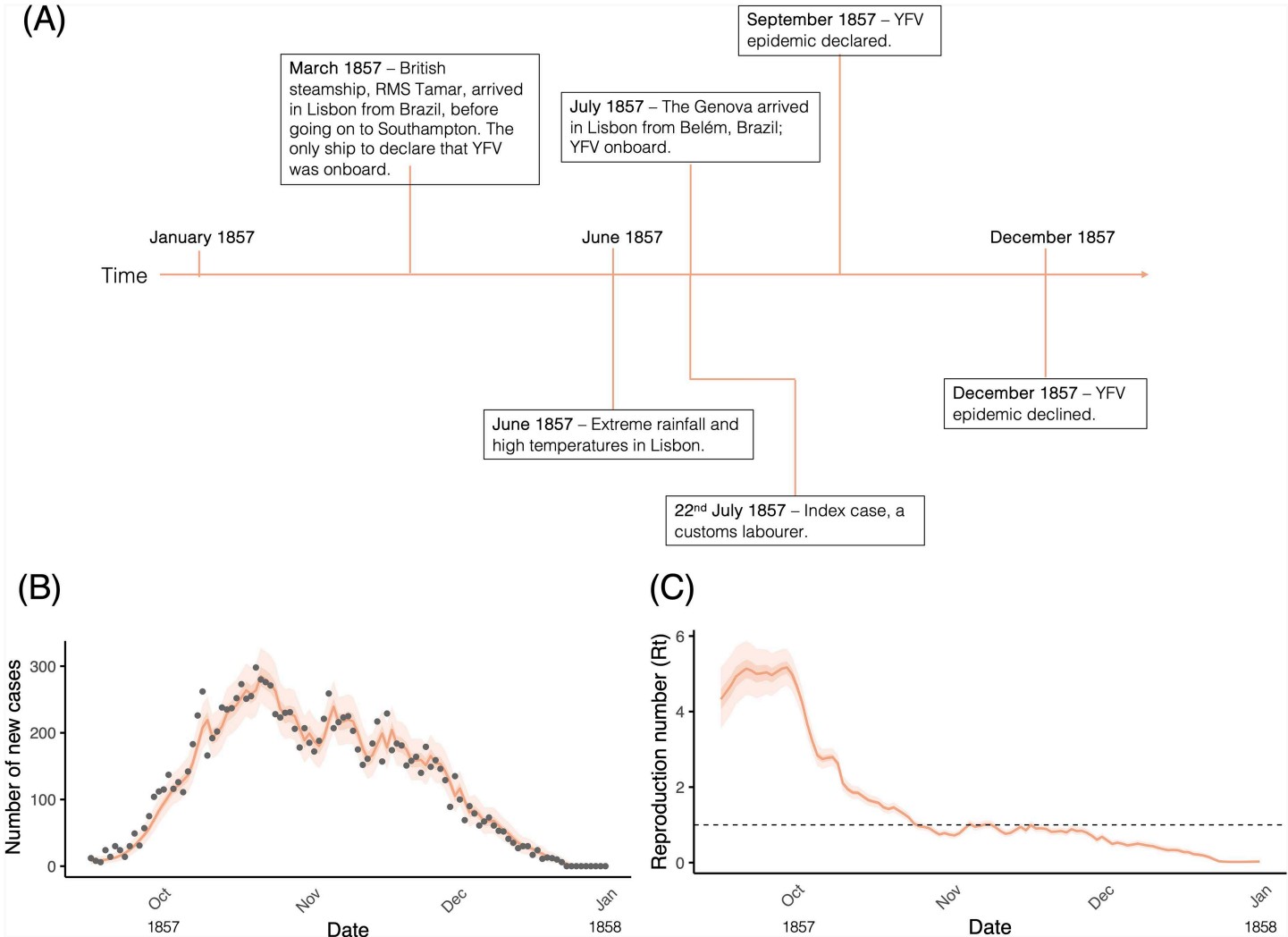

**Fig 1. Timeline and characteristics of the 1857 YF epidemic in Lisbon. (A)** A timeline illustrating the key events of the YF epidemic **(B)** Daily reported cases (grey points) and one-step-ahead incidence prediction (orange line) (see [45]) with 95% and 50% credible intervals (orange shading). **(C)** Rt estimates (orange line) with 95% and 50% credible intervals (shaded region) over time. The horizontal dashed line marks the epidemic threshold (Rt = 1).

ill with YF. In March 1857, the British steamship *RMS Tamar* arrived in Lisbon from Brazil, with its captain declaring the presence of YF cases onboard when it arrived [22]. According to reports, YF settled in Belém from 1850 for at least half a century, suffering 5,205 fatalities in this time, and YF spread from here to other Brazilian ports, such as Soure and Vigia [31], which demonstrates the infection potential from the city. We find parallels with cases of YF detected in 1850 Porto, when the merchant ship, *Duarte IV,* arrived from Brazil, entering the Douro River Valley where five customs guards fell ill with YF, killing three [22].

The first records of the YF Lisbon outbreak were reported less than a month after the extreme rainfall in June 1857. We found that June 1857 experienced the highest relative rainfall compared to the same month in 1855 and 1856, receiving 36.4 mm, much higher than the next highest (1.1 mm in June 1855). The driest months also coincided with the warmest months (See S3 Fig). Across the three years, the highest mean (24.0°C) and maximum (37.5°C) temperatures occurred

in July 1857. These conditions along with the poor sewer and drainage infrastructure in nineteenth century Portugal [61] created optimal conditions for *Ae. aegypti* breeding. During periods of intense rainfall, the existing sewer and drainage infrastructure was likely overwhelmed, leading to stagnant water pools that foster breeding habitats [62]. November 1857 experienced a particularly wet month, receiving 231 mm of precipitation, compared to only 4.2 mm of precipitation in November 1856. Further, temperatures began to decrease from the high averages observed in summer and autumn, dropping to an average temperature of 7.5°C in January 1858, and 11.3°C in February 1858. These conditions coincided with a decline in cases in all areas simultaneously from November onwards, until cases had disappeared entirely by February 1858.

## Epidemic characteristics

According to official records, the YF epidemic in Lisbon was associated with 13,757 cases and 5,652 deaths (CFR = 41%) in the period July 1857–February 1858. An epidemic was declared in September 1857. Our analysis of the basic reproduction number (Fig 1B) implies that without control in a fully susceptible population, each YFV infection transmitted by *Ae. aegypti* could lead to, on average, five secondary infections. Official records report that the index case was a customs employee that fell ill on 33 Rua da Padaria on 22nd July 1857, and died five days later [22]. The index case worked in the port, handling luggage and inbound ships' contents, which were likely harbouring infected vectors. The second case (29th July) was a neighbour of customs employees, and the third (1st August) was a person who lived in the same building as the index case.

Case incidence rose sharply, reaching a maximum of nearly 300 cases in mid-October, before decreasing steadily through the end of the epidemic (Fig 1B). Correspondingly, our analysis of the effective reproduction number illustrates a decrease from early October to $R_t$ around 1 by early November, dropping below 1 thereafter (Fig 1C). Since most fatalities occurred within the 34 main parishes of urban Lisbon, with an enumerated population of 133,677 in 1849 [44], we estimate that 4.2% of the population of urban Lisbon died of YF.

## Deaths by age, sex, access to healthcare, and neighbourhood

Out of the 5,652 officially reported deaths, 95.5% died at home or at one of six hospitals that specialised in the treatment of YF infections. Of the 5,398 deaths that occurred at home or at a specialised hospital, 64.2% of patients died at home and 35.8% of patients died in one of the YF-specific hospitals. An additional YF hospital was opened only for a few days and received few patients in Largo do Conde Barão, an area located in the Bairro Alto neighbourhood. Other non-specialised hospitals were pre-existing, permanent hospitals which were not specifically used for YF. These included military, navy, and private hospitals. Number of deaths at home and in specific hospitals are summarised in the S3 Table.

In terms of patient outcome, 44.2% of all patients that were treated at home died and 37.4% of all patients treated in YF-specific hospitals died. For deaths occurring at home, the 31–40 age group recorded the highest number of fatalities (675; 19.5% of home deaths), whereas the 21–30 age group was worst affected in YF-specific hospitals (585; 30.3% of YF-specific hospital deaths). The 21–30 age group accounted for the highest total of fatalities [1,127; 20.9% of all fatalities at home or YF-specific hospital (S4A Fig)]. The age group to have the highest fatalities per population of that age group was 51–60 in both males and females (Fig 2A). 17.0% of Lisbon's males and 11.5% of Lisbon's females aged 51–60 were killed by YFV. More deaths were seen at home than at YF-specific hospital for every age group in both males and females, except for males in the age groups 11–20 and 21–30, who suffered more deaths at hospital.

More men were admitted to hospitals for YF than women (see S4B Fig). The largest quantities of patients affected occurred in the peak of the epidemic in October, followed by a decrease in November. The Desterro hospital received the most patients (48.7% of all admitted to YF-specific hospitals) due to its increased capacity [22]. The peak admittance to hospital was 130 people on 20th October. A proportion of the population was reported to be effectively 'vaccinated' (S4C Fig). With the only vaccination in circulation in 1857 being for smallpox, and no YFV vaccination having been produced

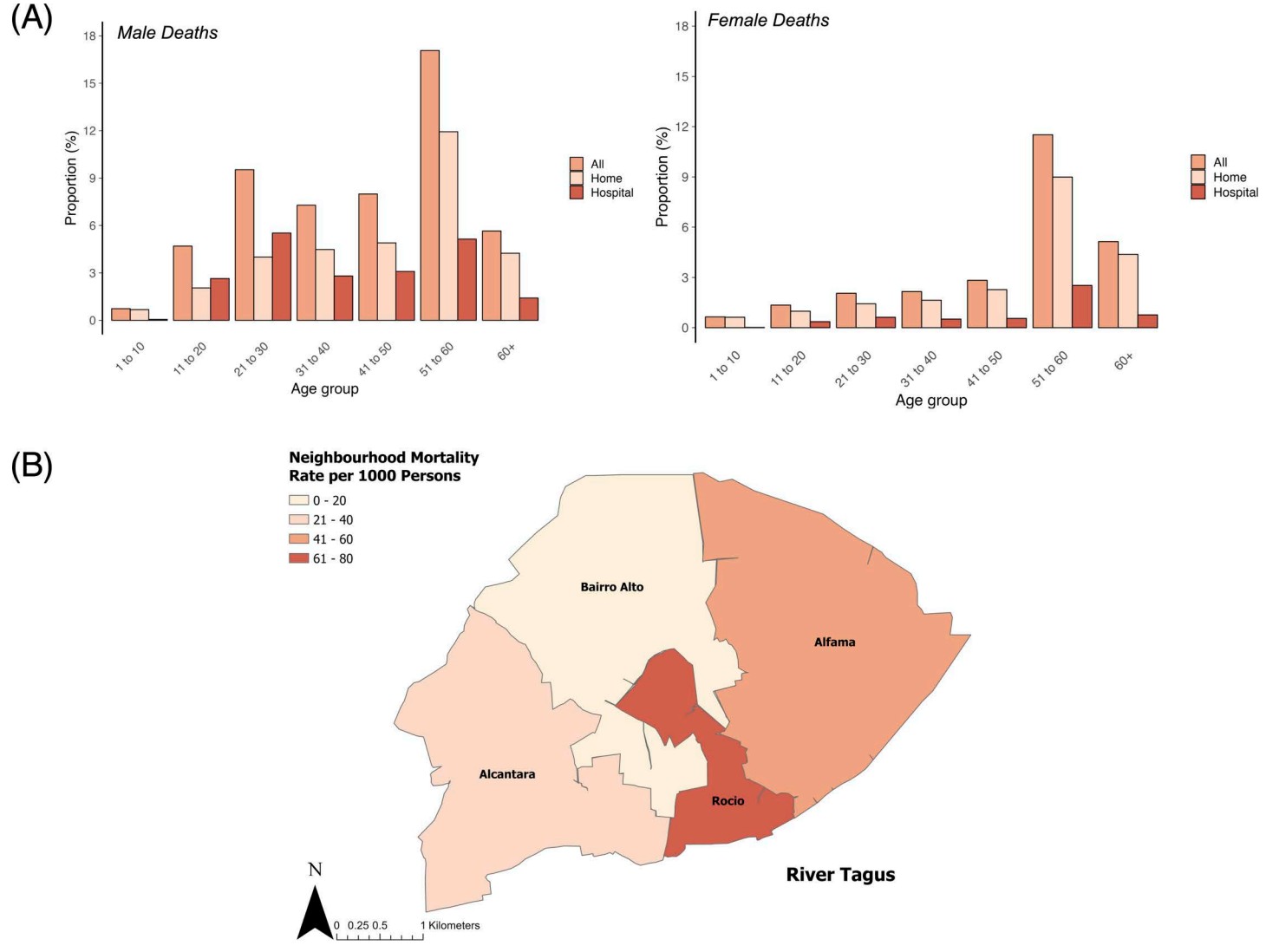

**Fig 2. YF mortality in Lisbon's populations during the 1857 epidemic. (A)** Deaths by age group and sex at either home or YF-specific hospital, as a proportion of total population of the respective age groups. **(B)** A map of the neighbourhood mortality rate per 1,000 persons. Digital neighbourhood boundaries created by the authors, based on the information in ATLAS Cartografía Histórica [52] and the work of Alves [40]. The authors have received permission to license the neighbourhood boundaries under CC BY-4.0 without restriction. The data that was traced over is available at: http://atlas.fcsh.unl.pt/cartoweb35/atlas.php?lang=en.

until 1937 [20], the 'vaccinated' were likely protected against smallpox. The mortality rate among the vaccinated (for small-pox) was 1:3.3, and among the unvaccinated, 1:2.5. Of the 1,932 hospital fatalities, 351 were vaccinated and 894 were not. The vaccination status of the remaining 687 had not been declared. More men than women were vaccinated.

The epidemic spread across all four neighbourhoods of Lisbon: Alfama, Rocio, Alcantara, and Bairro Alto (Fig 2B). Alfama experienced the highest fatality count with 36.7% of all fatalities that occurred at home or in one of the six YF-specific hospitals (1,982 fatalities). However, Rocio was the worst affected neighbourhood per head of population; 6.5% (1,781) of its population died (S4D Fig), or there were 65 fatalities per 1,000 persons. Bairro Alto was affected the least, with 20 fatalities per 1,000 persons. Males accounted for the highest proportion of deaths that occurred at either home

or YF-specific hospital (3,605; 66.8%). Women accounted for 388 hospital deaths, but only 1,118 women were admitted to hospital (hospital CFR = 34.7%), compared to 1,544 male deaths from 4,043 admitted (hospital CFR = 38.2%). Among women, there was a later decision to seek medical aid from hospitals [22].

## Death by occupation

Agriculture was found to be the least affected occupational group with just eight deaths (Fig 3), all of which were men. The most affected group was that of 'no designation' with 1,641 fatalities, mostly in women and occurring at home. The domestic occupational group formed the largest proportion of hospital fatalities. Many of the earliest cases were seen in customs employees.

## Mortality patterns at the parish and street level

All 34 parishes were impacted by YF (Fig 4A). Only 33 are detailed individually as the Health Council included the fatality count of the parish of S. Pedro within the total for the parish of Lapa (2). In descending order, the three parishes with the highest death tolls were Sé (32 – the first infected), Anjos (20), and Soccorro (19), each recording between 345 and 390 fatalities. S. Lourenço (26) was the only parish to have more deaths at hospital than at home. The highest parish mortality rates per 1,000 persons were Sé (32) with 213, Magdalena (33) with 115, and S. João da Praça (29) with 108 (Fig 4B). Our hot spot analysis identified clusters of high parish mortality rates across 33 parishes (Fig 4C). 13 parishes were identified as hotspots with 99% confidence level. These were the parishes where the highest number of deaths fatalities were clustered, towards the coastline and just east of central Lisbon. Two parishes were identified as a hotspot with 95% confidence level. These too were on the coastline and towards central Lisbon. One parish, Coração de Jesus (6), was identified as a cold spot with a 95% confidence level. This indicates the parish cluster with the lowest mortality rates. Parish

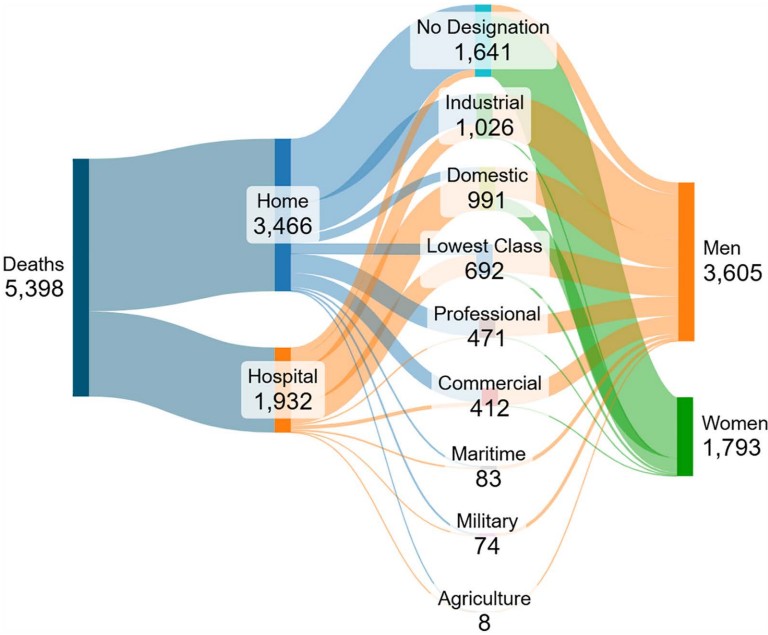

**Fig 3. A Sankey diagram delineating the proportion of all deaths occurring at home and in special hospitals in relation to occupational group and sex, during the 1857 YF epidemic in Lisbon.** The wider the flow, the higher the density. Source: based on data in Health Council ([22], tables 9 and 21, pp. 75, 99).

PLOS Neglected Tropical Diseases

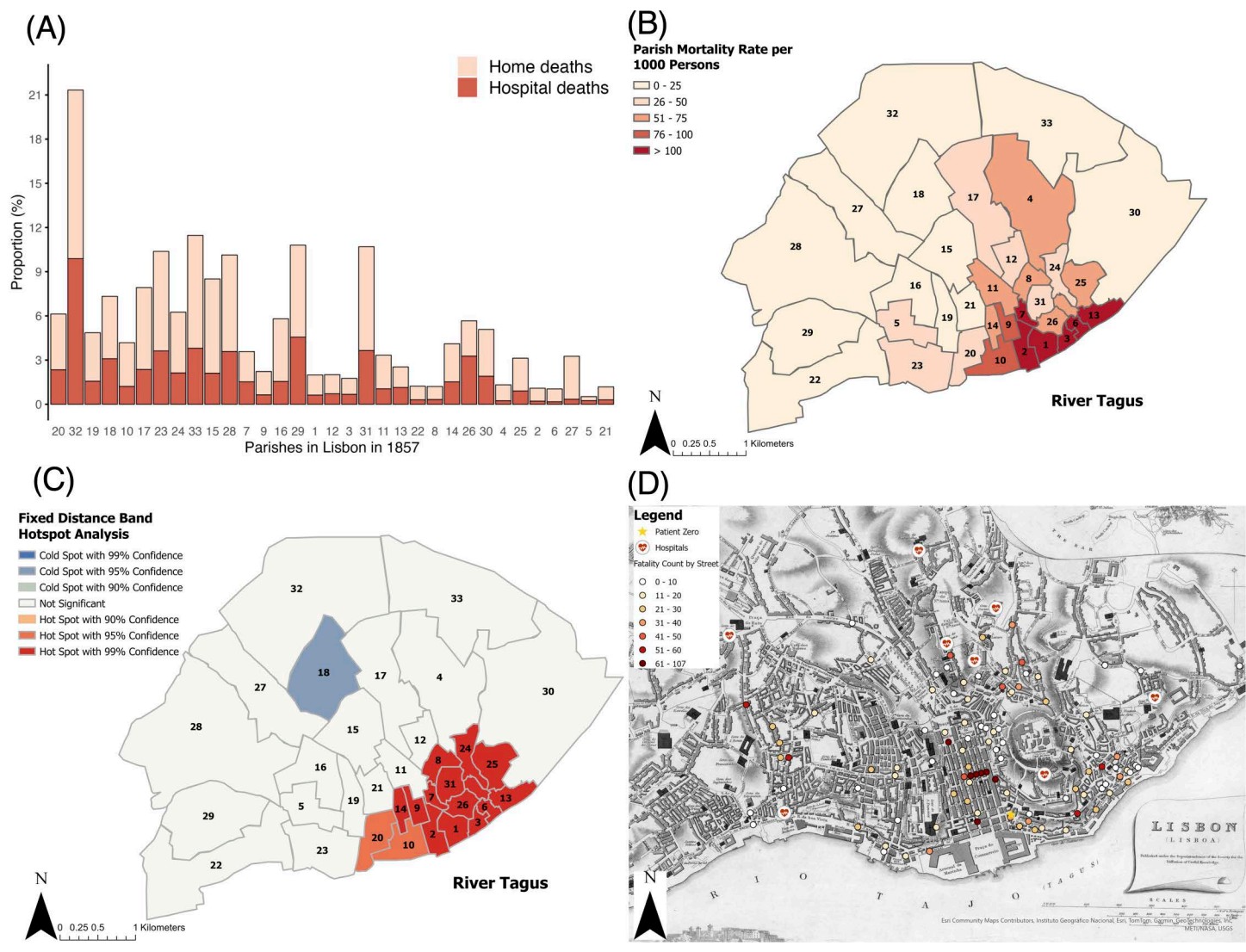

**Fig 4. YF mortality across Lisbon parishes and streets during the 1857 Lisbon epidemic. (A)** Clustered column chart of the distribution of fatalities at home and at hospital in each parish of Lisbon. **(B)** The mortality rate per 1.000 persons, by parish. The numbers within the parishes represent the order in which the epidemic progressed – number 1 represents the first infected, number 34 represents the last infected, based on reported data. **(C)** Getis-Ord Gi* hot spot analysis map, using fixed distance bands. The numbers within the parishes represent the order in which the epidemic progressed. Digital parish boundaries created by the authors, based on the information in ATLAS Cartografía Histórica [52] and the work of Alves [40]. The authors have received permission to license the parish boundaries under CC BY-4.0 without restriction. The data that was traced over is available at: http://atlas.fcsh.unl.pt/cartoweb35/atlas.php?lang=en. Map scales are 1:27,500. **(D)** The spatial extent of the epidemic on the street scale. 101 spots are applied to the centroids of streets. A star represents the residence of Jose Francisco, patient zero. Medical symbology represents the locations of the seven special YF hospitals and one alternative hospital – from left to right: Rua de Santo Ambrosio, Largo de Conde Barão, Rilhafolles, Calçada de Santa Anna, São José (alternative), Desterro, Largo dos Loyos, and Campo de Santa Clara. Historic base map drawn by Clarke [51]. Map scale is 1:15,000.

mortality rates per 1,000 persons ranged from 5 (S. Sebastião da Pedreira/5) to 213 (Sé/32). The mean number of deaths per parish was 160; 5,307 deaths were accounted for. The least affected parishes were found on the perimeter of the city.

We analysed mortality at the street level across Lisbon (Fig 4D). The street with the highest death toll was Rua da Prata with 107 fatalities, while Beco do Mexia had the fewest with only 3. Seven streets suffered more than 60 deaths, while the mean was 22 deaths per street. The mapped streets accounted for 2,240 deaths at home.

## Wind and elevation

On five separate occasions, winds blowing from NNW-NE direction (from inland; not from source at customs) coincided with lulls in the number infected [22]. This occurred on 10th October and the following days, at the end of October, 10th November, 20th November, and from 5th December until the end of the epidemic [22]. Patient zero was within the conservative buffer zone (533 m) representing the distance that mosquitoes can be blown from their source at customs (Fig 5A).

The highest fatalities per street are also seen to be in the lower lying areas, particularly in the valley just north of the coast (Fig 5B). The impact of elevation was seen on the intra-street scale, as well as on the city scale [22]. Streets with the highest fatality counts were found in low-lying areas on the coast and in the valleys (less than 30 m altitude; Fig 5C).

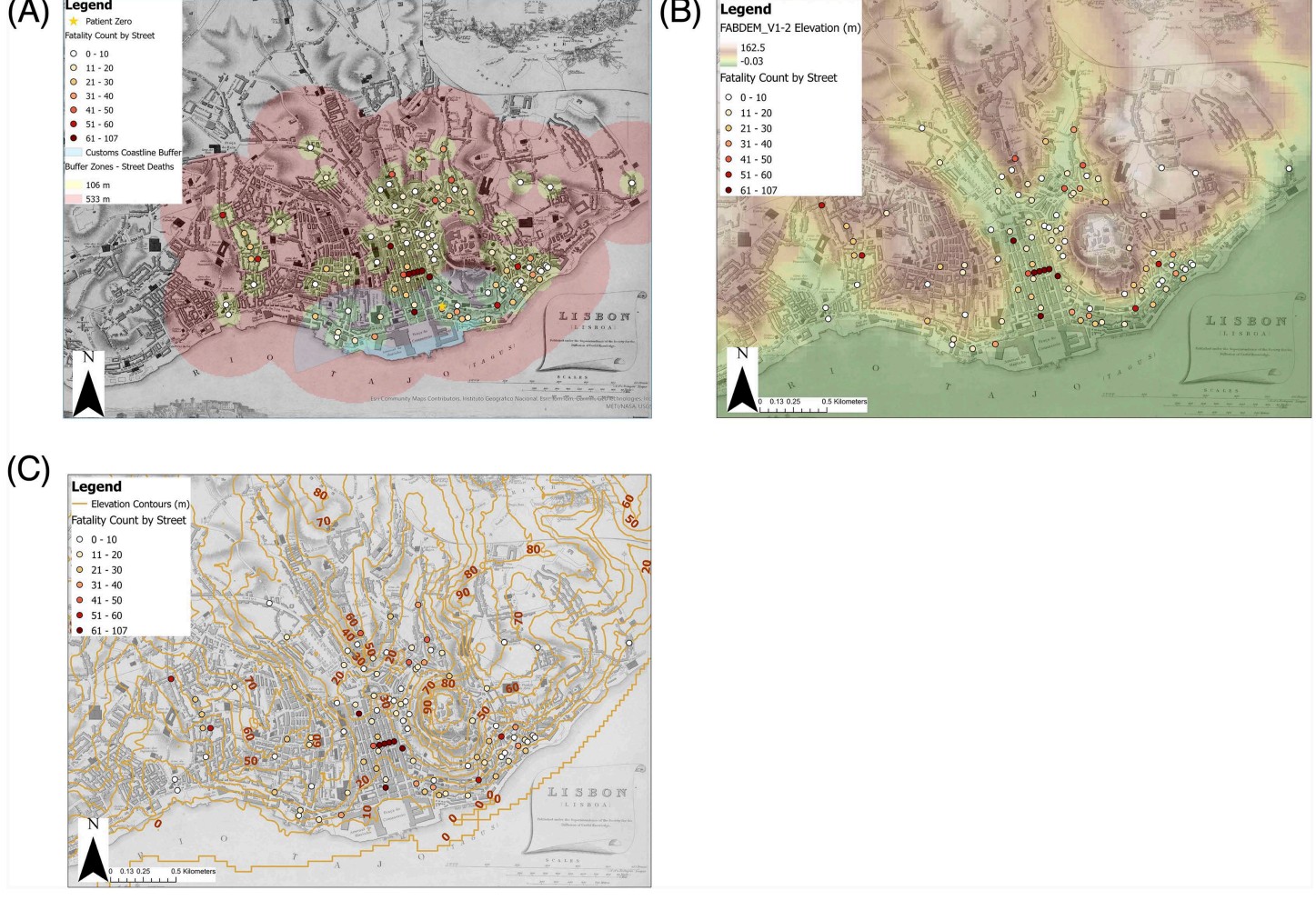

**Fig 5. YF deaths by elevation during the 1857 Lisbon epidemic. (A)** A map displaying multiple-ring buffer zones at 106 m (yellow) and 533 m (red) around each street spot and on the coastline at customs (blue) to investigate wind-blown vectors. The location of patient zero is included for context. Map scale is 1:17,500. **(B)** Dynamic Range Adjustment (DRA) Digital Terrain Model (DTM) and fatality count by street. **(C)** Elevation contours at 10-metre intervals derived from the aforementioned DTM, and fatality count by street. Map scale is 1:15,000. Elevation data source: Neal and Hawker [60]. The authors have received permission to license the digital elevation data under CC BY-4.0 without restriction. The data is available at, https://data.bris.ac.uk/data/dataset/s5hqmjcdj8yo2ibzi9b4ew3sn. Historic base map drawn by Clarke [51].

## Discussion

We combined historical YF case records with socio-environmental data to examine a major European outbreak in the pre-vaccination era, offering insights into how YF could re-emerge and spread in underexposed and unvaccinated populations. Our historical analysis provides a framework for interpreting YF transmission under comparable socio-environmental conditions today. This is particularly relevant amid concerns that YF could return to Europe as environmental conditions become more favourable and cross-continental travel from endemic regions increases in frequency and speed, raising the likelihood of importation [10]. By strengthening our understanding of how YF has previously interacted with Europe, we can improve preparedness and resilience to its potential re-emergence, and use historical evidence to inform future projections for *Ae. aegypti* suitability [34].

Franzini reported that 1857 had been a relatively 'healthy' year until August, with a lower mortality rate than previous years [22]. Our finding of the basic reproduction number ($R_0 \approx 5$). for the 1857 epidemic is aligned with the declaration of the YF epidemic in September 1857. We found that variation in $R_t$ corresponds to the progression of disease spread over time, peaking in October and declining in November and dying out by January. Our $R_0$ is consistent with the existing literature on $R_0$ for YFV. Liu & Rocklöv [63] reviewed YF papers published between 1950 – 2020 and found an average $R_0$ of 4.81. Within the papers reviewed, three studies looked at historic YF epidemics that occurred in New Orleans and Memphis Tennessee, USA in 1878. In these studies, the estimated $R_0$ ranged from 2.38–11. Our findings contribute to the growing literature on the reproduction number of YFV, including identifying changes in the time-varying $R_t$ throughout the course of the epidemic, predating mass vaccination.

Cases of infection began to decline after reaching a peak on 20th October, coinciding with the onset of colder and wetter months. While storing rainwater encourages vector breeding to an extent by creating stagnant water pools [62], there is less need for the long-term water storage which was used through the dry, summer months, and intense rainfall can wash away developing larvae from breeding sites [59]. November 1857, which saw a large decline in cases, was particularly wet. A decrease in average temperatures from their peak in July 1857 (24.0°C) will have also contributed to a decline in *Ae. aegypti* populations as temperatures moved further away from the optimal temperature for offspring production (29.2°C) [64]. Only temperatures above 12°C are suitable for population growth [65] and monthly average temperatures dropped considerably below this in December 1857 (9.8°C). The number of cases continued to decline in all locations throughout the winter months, before ceasing to occur in February 1858. A simultaneous decrease in infection on the city scale suggests that the change in environmental conditions was a strong driver of decline, removing the optimal conditions for *Ae. aegypti* breeding.

It is worth noting that there was a possible population-based reason for the epidemic's decline; by the end of November, the number of cases in the two worst affected parishes, Sé and Magdalena, had dropped drastically due to the high rate of population decline [22]. These parishes experienced the highest mortality rate so death and emigration had reduced the population vulnerable to new infections greatly [22]. Being in the optimal locations for YF, the new lack of infectivity potential will have caused a marked decline in overall infections.

Investigating the likely source of YF introduction to Lisbon in 1857 is of importance, providing insights into the mobility of the virus. While the *RMS Tamar* was the only ship to report YF cases in 1857, other 'suspicious' ships had been noted in the lazaretto (maritime quarantine first instated in Lisbon in 1837; see [32]). Although it was not possible to assign the cause to one ship due to many ships not reporting illness to avoid sanitary measures, competent *Ae. aegypti* were likely repeatedly introduced during this process [32]. YF was able to become established in Lisbon; the infection attack rate was at least 10.3% of the population, however it was implied that cases could have totalled closer to 18,000 across the course of the epidemic, reaching 13.5%.

Our maps show the highest mortality rates were in the neighbourhood of Rocio, the closest to the coast at the point where customs received infected ships, passengers, and cargo alike. This is where 13 parishes were identified to be hotspots of fatalities, across the neighbourhoods of Rocio and Alfama. The source of infection was at customs, dispersing

from the coast, which aligns with reports of YF primarily impacting port cities during the nineteenth century [66]. The 15 parishes with highest fatality rates were all part of the neighbourhoods of Alfama and Rocio, which were the most and least populous neighbourhoods respectively. Further, six of the seven most populous parishes had a fatality rate of 2.2% or less. While Rocio was the least populous neighbourhood in number of people living there, Rocio was more populous in terms of the number of people working there. Rocio is in low-lying, central Lisbon nearby customs and industries where many worked. For example, there was a high fatality rate of the commercial occupation group, such as traders, who frequented the customs offices [22]. Zahouli et al., [67] and Liu-Helmersson et al., [68] highlight that a higher population (density) would increase exposure to infected mosquitoes and transmission. Whilst true, this cannot fully explain epidemic extent in Lisbon 1857. The geographical distribution of the population is demonstrated to be an important driver of disease. Rocio and Alfama are located along the coast in central urban Lisbon, and spanned easterly, having a close relationship geographically, and socially (through workers living here), with the source of infection.

From our buffer analysis, patient zero lived within the more conservative buffer distance from the coastline at customs (506 m) and likely worked within the 106 m buffer from customs. This supports the view that YF entered Lisbon via customs, where contact with imported goods enabled local vector dispersal nearby. Most of the streets with high fatalities are within the closest proximity buffer of one another (106 m), and therefore, within the distance that mosquitoes were likely dispersed. To compound this, the Health Council reported that infection rates reduced when the winds were blowing from the north, towards the coast. This would suggest that the wind was hindering the dispersal of *Ae. aegypti* from the source at customs, after having been imported.

Proximity to the focal point of the epidemic was the greatest determinant of infection [22], as *Ae. aegypti* are easily dispersed. We found that most fatalities occurred within proximal streets where vectors are dispersed along connecting roads. Street fatality points fall within the closest buffer of each other (106 m), where the landscape is not flat or fragmented, illustrating how they are within adequate proximity for *Ae. aegypti* to be blown from street to street. Although *Ae. aegypti* are typically endophilic (rest and bite indoors), there are opportunities for the vectors to be involuntarily moved by wind as a passive transport mechanism when settled near homes, but outside; there are cases of *Ae. aegypti* being partially exophilic [69,70]. Further, the vectors must be dispersed from the source at customs after importation before they are able to circulate near buildings. While this could be from the transportation of water barrels or luggage with vectors harbouring inside, wind may also play a role in this dynamic. De Almeida [71] highlighted the role that streets may play in vector dispersal as they generate wind 'tunnels', funnelling mosquitoes into urban spaces. This is supported by Yamashita et al., [72] who demonstrated that wind assisted the urban distribution of *Ae. aegypti* in two Brazilian cities, and Marcantonio et al. (2019) [57], who concluded that wind contributed to involuntary movements of *Ae. aegypti* in the city of Madera, California. Although proximity to epidemic foci was significant, other drivers played a role in the transmission of YF given that proximity to hubs of cases in hospitals did not cause more damage [22] since only one died on Rua Largo do Desterro and two on Rua de Santo Ambrosio, indicating that proximity alone was not sufficient to cause epidemic damage. While wind facilitated vector dispersal, their establishment was likely also driven by high housing density and the abundance of stagnant water in these areas. We find that wind played a role in dispersing YFV during the 1857 epidemic but varied with elevation.

Elevation was also found to be related to the distribution of YF, not only indirectly through influencing the distance at which wind can disperse vectors but also acting as a natural barrier. Vectors do not travel far from their hatch sites without the influence of wind but can be barred from 'climbing' over higher elevations [59]. Our results suggest that the hills to the west of the city seemingly shielded the neighbourhood of Alcantara, leaving it far less affected than during past outbreaks of other diseases, such as cholera [22]. The hills to the west, dividing the centremost part of the city from Alcantara, provided a barrier to infection where conditions may otherwise have been optimal for transmission; Alcantara neighbourhood includes the three most populous parishes. Generally, we found that people living at higher elevations were at a lower risk of death from YF; few deaths occurred on streets with elevation above 80 m overall. This, however,

could have been due to lower population density at higher elevations. Though Santa Catharina, of higher elevation at 40 m altitude, still suffered more than São Paulo. Higher altitude areas would likely have had less movement of both people and vectors; individuals may have been infected in lower areas, and returned to higher areas to continue the transmission cycle. Additionally, the parish with the highest average altitude, Santa Cruz do Castello, only suffered 19 fatalities, but given its small population the fatality rate still reached 3.3%. Located near the coast and thus in an optimal location for YF exposure, Santa Cruz do Castello is a hotspot due to its surrounding parishes but was the 31st parish to be infected. This indicates a clear delay in the parish being impacted by the epidemic, despite being near the worst affected parishes that were infected during the early onset stages of the epidemic. The densely populated streets with less than 20 m in altitude were within proximity to customs, primarily housing customs employees, and were the worst affected streets. This aligns with the findings of Meena [73], who stated that lower-lying urban areas perpetuate the spread of *Ae. aegypti* mosquitoes, in part due to the accumulation of rainwater at low altitudes, and therefore vector breeding sites.This illustrates that wind and elevation may work synoptically in driving YF by determining the spatial variation in vector distribution.

Our findings on the impact of elevation and wind allows insight into how the environment may shape YF in present-day Lisbon. Lisbon's urban development is greatly dictated by its natural boundaries and geomorphological conditions [74]. The present-day parish of Marvila sits on the coast with significant elevation change, which results in a less compact urban morphology, with buildings irregularly covering the landscape [75]. In the context of YF's potential return, this parish may experience greater YF burden where elevation is lower and houses are more concentrated, while the more fragmented urban landscape means that there are weaker transport connections, slowing the movement of people [75], and vectors via wind tunnels. Further, the present-day parish of Santa Maria Maior, an amalgamation of the former historic centre that included the worst YF-affected parishes in 1857 (such as Sé, Magdalena, and São Miguel) has undergone mass touristification [76]. Along with the commodification of places of historic culture, this parish has been urbanised, particularly with short-term rentals [77,76]. Given the rate of urbanisation in locations evidenced to have optimal environmental conditions for YF, it can be inferred that similar urban transmission cycles could occur in the future, putting the same areas at risk.

Along with the mobility of vectors it is important to consider the mobility of individuals in contributing to the widespread transmission of YF. Places of congregation, such as taverns, churches, and schools, would have been suitable sites for *Ae. aegypti* to reside in or nearby, as they provide abundant opportunities for blood meals; this was demonstrated with the case of schools [78] and catholic churches in Merida City, Mexico [79]. In these locations, transmission occurs independent of vector mobility. Historically, and more comparatively to Lisbon in 1857, Sims [80] reports that 40–50% of 2,000 fatalities during the 1873 Memphis, USA, YF epidemic attended the same church (St. Brigid's Catholic Church). Though another driver may have been involved in causing so many fatalities in parishioners from this church, such as individuals living locally and close together, it is likely that the church acted as a hotspot for infections. The vectors were present in a public place where individuals congregate, providing a plentiful blood source. In nineteenth century Lisbon, taverns were a place where large groups would congregate, of which a large proportion were men; there were at least 1,556 'drinking establishments' in 1825, which illustrates the prevalence of this form of public space in Lisbon [81]. Further, boarding houses that primarily provided temporary or semi-permanent accommodation for men were often located towards the coast in the parish of Rocio [22], where the accommodation would be close to places of work, such as customs and industry. The aggregation of individuals in these boarding houses in the worst-affected neighbourhood, in an optimal geographic location for YF, may have acted as an opportunity for infection from vectors, as was observed during eighteenth and nineteenth century YF epidemics in New York, USA [82]. Given that men accounted for most cases and fatalities during the 1857 YF epidemic, it is likely that many of these infections occurred in androcentric places of congregation. Our research contributes to limited literature on the role of places of congregation in YF transmission. While contemporary studies mainly focus on YF transmission in rural or forested areas [83,84], our findings unveil how YF can spread in dense

public spaces and can inform vector control programs, focusing on public places, rather than the home. Future research efforts could map and analyse the spatial relationships between YF burden and places of congregation.

YF cases were mainly found where housing was most concentrated. Indoor-dwelling *Ae. aegypti* can thrive in homes, staying within 30–60 m of their hatch site in the absence of wind [59]. Homes in the urban area provided a source of standing water for vector breeding [85,86], since piped water was not made compulsory until 1872 in Lisbon. Many Portuguese households relied on traditional water supply in households or communities, comprising of springs, 6 wells, and 42 fountains [85,87], which can foster the breeding of *Ae. aegypti* near homes. Further, in 1850, some individuals bought their water at the house door from some 3,000 water carriers, who filled barrels at public fountains and delivered the water supply to houses [87]; this presented an opportunity for *Ae. aegypti* to be dispersed throughout the city, harbouring within barrels of water. A lack of drainage can lead to excess and stagnant water which can foster mosquito breeding. Better drainage can help limit mosquito breeding habitats [88,89,90].

We found that while being more affluent would have been beneficial in avoiding other diseases, such as living in cleaner areas during the 1856 cholera epidemic that killed 3,600 people in Lisbon [91], wealth offered no direct protection against YF. Desirable living conditions within the city offered no protection among the rich who still inhabited the worst affected places [22]. In 1857, the neighbourhood of Alcantara was not affected as badly as may have been expected (2.5% of population died; 828 deaths) despite being the poorest neighbourhood and usually dominating the fatality accounts of other diseases [22]. This is likely because Alcantara sits on the western periphery of Lisbon and was mostly agricultural land until the second half of the nineteenth century when it became increasingly industrial [74]. The Alcantara valley is separated from the city centre by a natural topographic barrier (hills), thus, mosquitoes favouring lower-lying areas would be less likely to breach the higher elevation between Alcantara and the rest of the city.

As reported by the Health Council, poorer populations living in worse conditions generally had a greater proportion of people treated at hospital than at home [22]. This suggested a social dynamic of who was treated and where. Traditionally, hospitals were designed for poor patients that could not be treated at home [91]. The most unscathed were those who worked in agriculture, as they lived and worked on the fringes of the city ([22]. Fewer buildings and lower population density in the agricultural fringes likely reduced *Ae. aegypti* activity and cases in these areas.

Only 2.4% of all deaths at home or YF-specific hospitals were in children aged 1–10 years, despite forming 14.4% of the total urban Lisbon population. This could have been due to children being too young to work on the ships, or in customs, and not being near the source of infection. However, they would have been just as at-risk of infection by being indoors at home. In fact, 94.7% of deaths in this age group occurred at home. Another reason for the small proportion of fatalities in children under 10 years of age is that in the nineteenth century, children were thought to be immune to YFV so any deaths from YF may have been misattributed to another cause of death. Certainly, this was common in nineteenth century YF epidemics in the USA [66], the highest fatalities were among men of working age.

Occupational groups are deeply rooted in historical gender disparities, which shaped the reporting of YF fatalities. Men accounted for at least 63.8% of all reported YF fatalities in 1857 Lisbon, despite men comprising only 44.8% of Lisbon's population. This is similar to the 1870 YF epidemic in Barcelona where men accounted for 62.1% of all deaths [27]. Most fatalities occurring at YF-specific hospitals in Lisbon were men assigned to the industrial, domestic, or 'lowest class' occupation groups. Out of the 1,026 fatalities in the industrial sector in 1857, just 25 were women. Earlier cases predominated in males of working age, such as patient zero who was a customs employee. The Health Council [22] remark on the 'professional' professions being associated with a high fatality rate in proportion to the total population employed in this sector; most of these were men that died at home. Professionals would reflect the more educated, which would be a result of, or result in, a higher socioeconomic status, and therefore the ability to afford treatment at home due to the higher salary associated with literacy [92]. Also, a high death rate in the professional occupations can be attributed in part to the high death rate in ecclesiastics (more than 30 deaths) [22]. A high fatality rate in ecclesiastics owed itself to contemporary

religious beliefs; clergy men would provide Christian spiritual care for patients and were at increased risk of infection during these activities.

While men accounted for most cases and deaths at both home and hospital admissions, there were still significant reports among women in 1857 Lisbon. Given that 77.0% of women belonged to occupations without a designation, it is likely that women undertook domestic duties at home. In Portugal, women had only started entering different sectors of employment at the end of the nineteenth century, and the 1867 Civil Code had declared them as still dependent on men, and they were viewed as minors [93]. The presence of women at home is supported by higher reports of death at home as opposed to hospitals. 7,842 people were treated at home; however, the report does not specify the sex divide.

Women may have faced higher exposure to YF as they often stayed at home for treatment rather than seeking hospital care. The 1859 Health Council noted women's reluctance to enter hospitals. This likely was resultant of women feeling ashamed to seek healthcare due to social taboos and stigma surrounding the lower social status of women, and gender power dynamics, meaning that women were dependent on the husband's permission and money to access healthcare [94]. This is still being observed in Ethiopia relating to five other neglected tropical diseases (NTDs), including arbovirus lymphatic filariasis which is also transmitted by *Ae. aegypti* [94].

These factors, combined with traditional gender roles and limited healthcare access, shaped the risk of YFV-related deaths among women in 1857 Lisbon. Historically, recognising women as a high-risk demographic group could have enabled more targeted, efficient interventions, potentially preventing many avoidable deaths. Since spaces in society - from workplaces to homes - have been and continue to be, highly gendered, this raises the idea that YF itself is a "gendered disease" [95]. Traditional gender roles affected the activities performed by men and women, influencing relative vulnerabilities to the disease and access to treatment. These features of gendered disease are still visible today in modern cases of arboviruses such as dengue [96], Zika [97], and chikungunya [98], highlighting the pressing need to address the gendered dimensions of arboviral diseases.

The Health Council reported the presence of a vaccine that, although deduced from a small sample size, seemed to prove somewhat effective in reducing fatalities. The Health Council is unclear as to the exact nature of the vaccine but, as no YFV vaccination had been produced until 1937 [20], the 'vaccinated' were likely protected against smallpox. Since vaccines were not freely available in Portugal until the National Vaccination Program began in 1965 [99], it is likely that vaccination in nineteenth-century Portugal required payment. Additionally, evidence suggests that more men than women were vaccinated, indicating that vaccine access may have already been a sociodemographic determinant of YF outcomes as early as 1857. Although it appears that the vaccine reduces mortality, our study suggests that it simply masks the real cause of reduced mortality, which is wealth and access. Those with the money to do so could remove themselves from the urban boundary, avoiding the reach of *Ae. aegypti*. In this sense, the "effect" of the vaccine may partly reflect social selection rather than protection alone. Nonetheless, in the broader context of nineteenth-century public health, these early reports were influential in shaping professional confidence that vaccination could be useful against other diseases at a moment when vaccine efficacy was actively debated [22]. This finding also points to an ongoing challenge in public health. During arbovirus outbreaks, intervention effectiveness can be overstated if analyses do not adequately address confounding and inequities.

Our study offers a uniquely detailed insight into the impact of YF in the absence of a vaccine and illustrates how social (e.g., gender, living spaces, water supply and drainage, and occupations) and environmental factors (e.g., wind, elevation, temperature, and precipitation) drive outbreaks, working synoptically. The particular case of wind as a factor has been explored in terms of distance within an urban and historic setting, which adds to our understanding of the influence of wind on the transmission of YF, contributing to limited research on the topic. Further, examining an epidemic in a non-endemic region provides a strong example of the mobility of YF, and how vectors can migrate over distances much further than nature would allow, and still become established. This assessment is particularly imperative amid current discourses that YF has the potential to return to Europe as environmental conditions become more favourable and cross-continental

travel increases in frequency and speed from endemic regions, increasing the chance that YF is imported to Europe [10]. Increasing our understanding of how YF has previously interacted with Europe means that we can better prepared and more resilient to YF's return; the use of historical analyses can be applied to future projections, identifying areas likely to become suitable for *Ae. aegypti* establishment [34].

Our study is limited by reporting bias and the reliability of historical records, since the environmental and health data collection methods of the time were rudimentary. Indeed, the numbers relating to the total 7,842 people treated at home in 1857 Lisbon should be considered with care as data prior to 15[th] September is missing and some volunteers did not report their work [22]. Further, the Health Council stated that a lack of information on the social conditions and demographic structure of the population was proving problematic when investigating diseases in all major cities in the nineteenth century. To address this issue, we sourced an historic dataset (1849 census; [44]) independent of the Health Council, on Lisbon's demographic structure. In so doing, we recognise that changes to the size and structure of Lisbon's population may have occurred in the eight years preceding the 1857 epidemic. In addition, owing to a lack of concordance between the available epidemiological and demographic data, statistical interpolation and extrapolation were used to determine population counts from which fatality rates could be constructed. All rates and associated findings should be interpreted with these data caveats in mind.

## Conclusion

By digitalising and mapping archival disease data, we demonstrated that the environment, combined with sociodemographic factors, contributed to the progression of the 1857 YF epidemic in Lisbon. Aligned with previous studies, our estimates of $R_0$ illustrates that YFV is capable of rapid spread and decline predating vaccination. While the 1859 Health Council in Lisbon identified wind direction as the primary driver of YF, our findings highlight the complex interaction of human-environmental relations in shaping the epidemic. Out of the 34 parishes in urban Lisbon, our hot spot analysis illustrated a cluster of 15 high-risk parishes near the coastline. We found that the highest number of deaths occurred within connected streets confined in low-lying, built-up areas, illustrating the potential role of wind aiding mosquito distribution and concentration of mosquitoes near homes. Considerations of the role of places of congregation on YF transmission inform vector control programs, illustrating the transmission potential in public spaces, rather than predominately homes. Gender disparities existed in the reporting of YF fatalities. Although working-aged men accounted for most YF fatalities, the highest probability of death was among "non-designation" occupation individuals, mainly women working at home. Examining a historical urban epidemic in Europe provides a unique insight into how re-emergence of YF in Europe may look amid climate change and growing interconnectivity. This study demonstrates the complexity of YFV transmission in a historic urban context and illustrates the need for continued research to understand YFV transmission and its spatial dispersal.

## Supporting information

**S1 Table. Referring to the map in S1A Fig, a list of the mapped parishes of 1857 Lisbon, indicating the parish name that corresponds to each number label in the map.**
(XLSX)

**S2 Table. The nine groups of occupations, as devised by the Health Council ([22], Table 21, p.99).**
(XLSX)

**S3 Table. A breakdown of the 5,652 deaths that were officially recorded during the 1857 Lisbon epidemic.** Data from Health Council ([22]: p.30).
(XLSX)

**S1 Fig. Reconstruction of mid-nineteenth century administrative geography in Lisbon.** (A) Digitised mid-nineteenth century Lisbon neighbourhood boundaries. (B) Digitised mid-nineteenth century Lisbon parish boundaries located on the bank of the river Tagus. Each parish is numbered – S1 Table provides information on the name of the parish that each number corresponds to. Map scales are 1:27,500. Digital neighbourhood and parish boundaries created by the authors, based on the information in ATLAS Cartografía Histórica [52]. (C) A timeline of the changing municipal organisation of Lisbon [40].
(TIFF)

**S2 Fig. Snippet of map (1of 6) of Lisbon's parishes used to trace the research's parish shapefile [52].**
(TIFF)

**S3 Fig. Monthly comparison of temperatures and precipitation in Lisbon for the year of the epidemic and the two years prior to identify potential environmental drivers of YFV.** (A) Three-year monthly temperature mean comparison (B) Three-year monthly temperature maxima comparison (C) Three-year monthly total precipitation in Lisbon comparison and (D) Relationship between monthly temperature means and total precipitation throughout 1857. Source: data from Royal Observatory, in Lyons [42].
(TIFF)

**S4 Fig. Health outcomes, including vaccination status, by age and sex.** (A) Total deaths by age group at either home or YFV-specific hospital, as a proportion of total deaths in these locations. Age-related information was unavailable (or 'undefined') for 0.8% of fatalities at home and 1.4% of fatalities at YFV-specific hospitals. (B) A comparison of admitted, cured, and dead at hospital by sex, with a temporal dimension. (C) The effectiveness of the vaccine, by sex. (D) The distribution of fatalities at home or YFV-specific hospital, by sex and neighbourhood.
(TIFF)

## Author contributions

**Conceptualization:** Isaac H. Bates, Sabrina L. Li, Nuno R. Faria.

**Data curation:** Isaac H. Bates, Kris V. Parag.

**Formal analysis:** Isaac H. Bates, Sabrina L. Li, Kris V. Parag.

**Funding acquisition:** Sabrina L. Li.

**Investigation:** Isaac H. Bates, Sabrina L. Li, Kris V. Parag, Matthew Smallman-Raynor.

**Methodology:** Isaac H. Bates, Sabrina L. Li, Kris V. Parag, Katy AM Gaythorpe, Matthew Smallman-Raynor, Nuno R. Faria.

**Project administration:** Isaac H. Bates, Sabrina L. Li.

**Resources:** Sabrina L. Li, Kris V. Parag, Nuno R. Faria.

**Software:** Isaac H. Bates, Sabrina L. Li, Kris V. Parag.

**Supervision:** Sabrina L. Li, Matthew Smallman-Raynor, Nuno R. Faria.

**Validation:** Isaac H. Bates, Sabrina L. Li, Kris V. Parag.

**Visualization:** Isaac H. Bates, Sabrina L. Li.

**Writing – original draft:** Isaac H. Bates, Sabrina L. Li, Matthew Smallman-Raynor.

**Writing – review & editing:** Isaac H. Bates, Sabrina L. Li, Kris V. Parag, Katy AM Gaythorpe, Ana B. Abecasis, Matthew Smallman-Raynor, Nuno R. Faria.

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
