## [Decision Letter · Decision Letter 0]

19 Nov 2025

Spatial and Social Determinants of the 1857 Yellow Fever Epidemic in Lisbon

Dear Dr. Li,

Thank you for submitting your manuscript to PLOS Neglected Tropical Diseases. After careful consideration, we feel that it has merit but does not fully meet PLOS Neglected Tropical Diseases's publication criteria as it currently stands. Therefore, we invite you to submit a revised version of the manuscript that addresses the points raised during the review process.

Please submit your revised manuscript within by Jan 18 2026 11:59PM. If you will need more time than this to complete your revisions, please reply to this message or contact the journal office at plosntds@plos.org. Please include the following items when submitting your revised manuscript:

We look forward to receiving your revised manuscript.

Kind regards,

Álvaro Acosta-Serrano

Section Editor

Shaden Kamhawi

co-Editor-in-Chief

Paul Brindley

co-Editor-in-Chief

**Journal Requirements:**

At this stage, the following Authors/Authors require contributions: Isaac Bates, Sabrina Li, Kris Parag, Katy Gaythorpe, Ana Abecasis, Matthew Smallman-Raynor, and Nuno Faria. Please ensure that the full contributions of each author are acknowledged in the "Add/Edit/Remove Authors" section of our submission form.

- © on page: 16.

5) We have noticed that you have uploaded Supporting Information files, but you have not included a list of legends. Please add a full list of legends for your Supporting Information files after the references list.

Potential Copyright Issues:

i) Figures 2B, 4B, 4C, 4D, 5, S1, and S2. Please (a) provide a direct link to the base layer of the map (i.e., the country or region border shape) and ensure this is also included in the figure legend; and (b) provide a link to the terms of use / license information for the base layer image or shapefile. We cannot publish proprietary or copyrighted maps (e.g. Google Maps, Mapquest) and the terms of use for your map base layer must be compatible with our CC BY 4.0 license.

7) In the online submission form, you indicated that The data in this study are available from the corresponding author upon reasonable request.. All PLOS journals now require all data underlying the findings described in their manuscript to be freely available to other researchers, either

1. In a public repository

2. Within the manuscript itself

3. Uploaded as supplementary information.

8) Please amend your detailed Financial Disclosure statement. This is published with the article. It must therefore be completed in full sentences and contain the exact wording you wish to be published.

2) If any authors received a salary from any of your funders, please state which authors and which funders..

9)  Please ensure that the funders and grant numbers match between the Financial Disclosure field and the Funding Information tab in your submission form. Note that the funders must be provided in the same order in both places as well.

**Reviewers' Comments:**

Reviewer's Responses to Questions

**Key Review Criteria Required for Acceptance?**

**Methods**

-Are the objectives of the study clearly articulated with a clear testable hypothesis stated?

-Is the study design appropriate to address the stated objectives?

-Is the population clearly described and appropriate for the hypothesis being tested?

-Is the sample size sufficient to ensure adequate power to address the hypothesis being tested?

-Were correct statistical analysis used to support conclusions?

-Are there concerns about ethical or regulatory requirements being met?

Reviewer #1: a) Objectives are well articulated with a clear testable hypothesis.

b) The study design was suitable elaborated to reach the stated objectives,

c) The population was described appropriately considering the historical records,

d) As far as authors were aware, the estimated sample was acceptable to address the hypothesis,

e) I am not familiar with all the statistical techniques the authors used, however they are well referenced.

f) ethical requirements were met.

Reviewer #2: Yes,

Reviewer #3: I believe the objectives were proposed in alignment with the availability of data and the possibilities for analyzing them, and the study was designed to address these objectives. The population is clearly described.

Reviewer #4: The proposed objectives are clearly articulated at the end of the introduction.

Regarding the hypothesis to be tested, it appears that the authors posited that the apllication of modern spatial epidemiological methods to historical data would enable a deeper understantading of past and current wellow fever virus infection within the framework of human-environment interactions.

The population affected by yellow fever, as well as the cases extracted from the 1859 report of the Extraordinary Council of Public Health of Portugal,is clearly described in the Materials and Methods section. It is also clear where the environmental and socioeconomic data were obtained from.

The statiscical power of the sample is inherently limited by the historical data available for the analysis conducted. And yes, the statistical inference supports the conclusion.

All items in the Materials and Methods section are well described an clearly explained. The supplementary figures and tables ensure a clear understanding o the study area and the census data.

Reviewer #5: The objectives of the study are clearly articulated. The methodology and analysis adequately described. No ethical or regulatory requirements was needed as the study is based on archived documentations.

**Results**

-Does the analysis presented match the analysis plan?

-Are the results clearly and completely presented?

-Are the figures (Tables, Images) of sufficient quality for clarity?

Reviewer #1: Yes, the analysis presented match the analysis plan.

Results were clearly and well presented,

The figures are of sufficient quality and clearly shown.

Reviewer #2: Yes,

Reviewer #3: Results are clearly presented and match the methods, and they are sufficient to answer the objectives.

Reviewer #4: The results are consistent with the proposed objectives. The figures, which include graphs, diagrams, maps, as well as the information from the supplementary material. are clear and highly valuable for understanding the results present.

The authors appear to discuss the results lines 283-285, 288-290, 305-307, 323-324, 380-381, 421-423, 445-447, and 453-456. I suggest that these observations would be more appropriately presented in the Discussion section.

On line 299, the authors should specifiy the species Aedes aegypti rather than just the genus. The genus includes other species and does not accurately represent the vector species.

Reviewer #5: The results presented match the analysis plan.

**Conclusions**

-Are the conclusions supported by the data presented?

-Are the limitations of analysis clearly described?

-Do the authors discuss how these data can be helpful to advance our understanding of the topic under study?

-Is public health relevance addressed?

Reviewer #1: The conclusions are supported by records from the nineteen century in other cities.

Limitations of the study are well reported, the reliability of historical records were affected by the lack of data collection methods at the time and that should have influenced the epidemiological analysis.

The authors considered these data are helpful to improve the comprehension of yellow fever dynamics.

Public health is considered through the records of Public Health Council. These records contributed much to describe the yellow fever outbreak in 1857 in Lisbon.

Reviewer #2: Yes,

Reviewer #3: Not entirely, but I suggest that the authors revise their conclusions to better align them with the reviewers' suggestions, should they decide to incorporate those suggestions.

Reviewer #4: The conclusions are supported by the results, and the study limitations were clearly reported in lines 636 to 650 in the Discussion section.

The study presented is centainly relevant to Public Health,particularly regarding the transmission of urban yellow fever.

I did not find in the discussion a clear explanation of how the data can be usefull for advancing our understanding of the tropic under study. Except for its contribution to the growing body of literature on the reproduction number of the yellow fever virus including the identification of chances in the Rt variable over time during the course of the epidemic.

Basead on the results obtained and comparisons with other studies, I ask ifthe the authors could discuss how wind and altitude may currently influence the dispersal or the vector and the urban yellow fever virus within the same neighborhoods in lisbon Portugal.In other word, such an internal extrapolation could be considered in light of present urbanization patterns and environmental conditions.

Reviewer #5: Overall the conclusions are supported by the presented data. However as mentioned in the section below more emphasis and clarification are needed on certain aspects

**Editorial and Data Presentation Modifications?**

Reviewer #1: No suggestions.

Reviewer #2: (No Response)

Reviewer #3: I will highlight here the suggestions and questions that are more specific to certain sections of the text, and in the following box some more general comments and suggestions. I would like a response from the authors regarding each of my comments:

1) Line 46 – Please clarify the statement: “...Aedes (Stegomyia) aegypti, which can circulate in both rural and urban areas (Gabiane et al., 2022).” I believe this is a generalization that leaves room for doubt regarding what exactly is meant by “circulation of A. aegypti in rural areas.” For example, this does not seem applicable to the situation of yellow fever circulation in the Americas. Do the authors have a different view? Could they comment/clarify?

2) Line 48 – Have the authors considered adopting the new designation “Orthoflavivirus,” according to Postler TS, Beer M, Blitvich BJ, et al. Renaming of the genus Flavivirus to Orthoflavivirus and extension of binomial species names within the family Flaviviridae. Arch Virol. 2023;168(9). doi:10.1007/s00705-023-05835-1, or is some other option being evaluated?

3) Line 59 – If applicable, I suggest using the original reference: Bryant JE, Holmes EC, Barrett ADT. Out of Africa: A molecular perspective on the introduction of yellow fever virus into the Americas. PLoS Pathog. 2007;3(5):0668-0673. doi:10.1371/journal.ppat.0030075

4) Line 61 – “YFV epidemic,” when the correct term should be “YF epidemic.” Please check the rest of the text, as this occurs repeatedly.

5) Lines 61–62 – Please check whether the following references might be earlier and therefore more appropriate:

• Bryant JE, Holmes EC, Barrett ADT. Out of Africa… (2007)

• Klitting R, Gould EA, Paupy C, de Lamballerie X. What does the future hold for yellow fever virus? (I). Genes (Basel). 2018;9(6). doi:10.3390/genes9060291

6) Line 74 – Do the authors consider the 1856 Lisbon epidemic to be a separate epidemic? What comments could they offer about the immune individuals generated by that epidemic and their relevance (or lack thereof) during the 1857 epidemic?

7) Lines 78–80 – I suggest strengthening the justification for choosing this particular outbreak. If the cited reference provides more data on magnitude, for example, it would be important to mention this.

8) Lines 82–83 – What is the rationale behind this statement? Did merchant ships departing from Brazil facilitate new introductions of the virus into Lisbon? This is not clear.

9) Lines 242–243 – The “analysis” at the time suggested a relationship between wind (or wind direction) and the spread of cases. Keep in mind that this “conclusion” was likely based on the idea of miasmas and not on vector-borne transmission. Where is this historical “analysis” linking wind described? Considering that Aedes is a mosquito that lives indoors, how do the authors envision such strong influence of wind?

In some situations, wind has been considered important for the dispersal of sylvatic vectors involved in yellow fever transmission in the Americas. I do not believe this hypothesis can be directly applied to urban transmission by Aedes, particularly when analyzing data from an epidemic that occurred 168 years ago.

10) Lines 440–447 – It does not seem that the data collected provide enough support to conclude that viral dispersal occurred primarily due to mosquito movement rather than the movement of infected people. Furthermore, the role of places of congregation—such as churches, public areas, taverns, and other meeting points—was not adequately addressed. Although these were included in the analyses, it seems the authors treated these locations as dispersion points from which infected mosquitoes were transported to other areas, rather than as venues visited by people who could become infected there.

11) Lines 499–500 – As highlighted by the authors themselves, I believe this point is debatable and deserves further discussion.

12) Lines 502–511 – Please carefully review the reference “Almeida, M. A. B., dos Santos, E., Cardoso, J. d. C., da Silva, L. G., Rabelo, R. M., and Bicca-Marques, J. C. (2019). Predicting Yellow Fever Through Species Distribution Modeling of Virus, Vector, and Monkeys. EcoHealth. 16: 95–108.” for two main reasons:

1. The cited study was conducted with sylvatic vectors in southern South America, in natural (non-urban) environments, and using contemporary data from the time of the event. The comparison does not seem directly applicable.

2. Note that the full text of the cited article states:

“However, we cannot reject the possibility that roads play a role in the dispersal of infected vectors either via transport in vehicles or the influence of wind ‘tunnels.’”

Yet the manuscript cites it as: “De Almeida (2019) highlighted the role that streets play in vector dispersal as they generate wind ‘tunnels’, funnelling mosquitoes into urban spaces. While wind facilitated vector dispersal, their establishment was likely also driven by high housing density and the abundance of stagnant water in these areas.”

13) Lines 513–533 – There may be differences in vector density and perhaps less human movement due to the topography (hills). In other words, higher-altitude areas might have fewer mosquitoes and fewer people moving around; individuals may have been infected in lower areas (where rainwater accumulates), and then, returned to higher areas where they could infect mosquitoes already present there, after teh incubation period.

14) Lines 535–537 – I consulted the article by Newman et al. (2024) (although I did not read it thoroughly, so I may have missed something), and I did not find this statement directly: “staying within 30–60 metres of their hatch site in the absence of wind.” Could the authors please clarify?

15) Lines 660–662 – I would like to know more about this conclusion:

“We found that the highest number of deaths occurred within connected streets confined in low-lying, built-up areas, illustrating the potential role of wind aiding mosquito distribution and concentration of mosquitoes near homes.” Once again, Aedes aegypti is a mosquito that lives indoors. It does not need wind to remain in these environments. When food sources (people) and oviposition sites (small water collections) are available, mosquitoes remain in place. Furthermore, I insist: what evidence do we have about the influence of wind inside homes in terms of carrying mosquitoes over certain distances?

Reviewer #4: I recommend a less thorough review for approval, in accordance with the observations made in the aforementioned items.

Reviewer #5: Here is an excellent study using archive data to recreate a historical or deathly epidemic. The depth and breath of the analysis and methodology is worth appraising and congratulating. However, there are, minor issues that need to be clarified prior to publication.

Overall, there is a lot of emphasis on the factors contributing to spread it will also be interesting to highlight if there are any factors that led to a decline of the epidemic

Secondly, contrary to available evidence, the authors highlighted in Line 478-492 that population density was not the primary driver of disease. Furthermore, in line 478-486. The least populated Rocio neighbourhood recorded the highest mortality rate. Was the Rocio neighbourhood least populous in terms of persons leaving in the area? Could it have been populous in terms of number of people working in the area?

Thirdly, line 159-160. Data on temp and precipitation were also retrieved, briefly mentioned in the results section but not discussed in great lengths as observed with the wind data. Considering the contribution of temperature is one of the study objectives, I will suggest for the inclusion of FigS3 as a main figure rather than a supplementary figure.

Minor

1) Abstract: Line 18 rephrase Despite the availability of a highly effective vaccine, …

2) Fig 1a – rephrase the statement “ The only ship to declare that YFV was on board.” I assume these are YF patients or suspected patients.

Line 282…declaring the presence of YF cases or patients onboard

Line 283 verify YFV in 1857 = report YF cases

3) Line 328 Figure 1 rephrase legend” Timeline and characteristic of the 1857 Yellow fever epidemic. It will be great to also consider the reconsider the legend of the other figures precising it’s the 1857 Yellow Fever epidemic

**Summary and General Comments**

Reviewer #1: The manuscript's title summarize the content and uses keywords to emphasize the occurrence of yellow fever outbreak in the portuary city, Lisbon, in urban space. As commented before to authors, lack of data collection methods should influence the epidemiological analysis. However, the discussion shows very support material and evidence arguments to demonstrate the dynamics of yellow fever at that that period at the city of Lisbon. There are no concerns about dual publication, research or publication ethics.

Reviewer #2: Re: Spatial and Social Determinants of the 1857 Yellow Fever Epidemic in Lisbon

Bates et al. conducted a digitisation of archival data and evaluated the influence of environmental and social factors on the yellow fever outbreak of that period. The study presents an engaging and insightful narrative that effectively demonstrates how digitising historical records can enhance our understanding of past public health events and yield valuable lessons for present and future disease control efforts. I read the article with great interest and found it both informative and enjoyable.

Discussion: This section is heavily dominated by the presentation of detailed historical data rather than by synthesising the key insights. It would benefit from a broader interpretation that highlights the main lessons learned and outlines potential strategies or scenarios to prevent the large-scale mortality and morbidity observed at that time.

Reviewer #3: I would like to commend the initiative taken by Dr. Li and her colleagues, which, in my view, represents an act of courage given the magnitude of the challenge. I consider this study to be a highly relevant effort by the authors. Revisiting historical yellow fever outbreaks and analyzing them in light of current knowledge is a very pertinent idea, one that may be necessary under certain circumstances and represents a creative way of reconstructing the history of the disease. This approach certainly required substantial research effort and meaningful collaborative work among the teams involved, especially considering the scarcity of data and information that characterizes certain historical periods—particularly when compared to the wide availability of data we have today. In addition to this scarcity, it is reasonable to assume that data accuracy was also lower compared with our current capacity to collect and consolidate data for subsequent analysis.

I would sincerely like to see a more consistent justification for choosing this specific outbreak. I did not find (or at least not in sufficient detail) in the article which factors led the authors to apply this proposal to the 1857 yellow fever outbreak in Lisbon, rather than to another similar event. It would be important for these justifications to be explained more thoroughly in the text.

An additional suggestion would be to consult the book "A história da Febre Amarela no Brasil", authored by Dr. Odair Franco (available at: https://bvsms.saude.gov.br/bvs/publicacoes/0110historia_febre.pdf), which presents data on historical outbreaks in Brazil and other countries, including a description of the measures adopted to contain the first recorded outbreak in Brazil, in the city of Recife. These elements could enrich the historical review presented in the manuscript.

I had the impression that the manuscript did not sufficiently address the potential for generating new cases in places of congregation, such as churches. The analysis seems biased toward the hypothesis that wind was the main — or one of the main — explanations for the spread of the virus by transporting infected mosquitoes. The contributions of human movement during viremia and the role of mild or asymptomatic infections seem underexplored, and I believe these aspects deserve greater attention. If the vector within the household (or very close to it) finds all the conditions necessary for its maintenance, why would mosquitoes seek other locations? Thus, it does not seem necessary for urban mosquitoes to move in search of more favorable environments. If we then imagine a “passive” transport mechanism such as wind, how would this operate indoors?

Similarly, I would like to see more comments on earlier outbreaks and their potential to generate immune individuals who would have been “protected” during the 1857 circulation of yellow fever. The text cites the arrival of yellow fever brought by the RMS Tamar in March 1857, but this seems insufficiently explored, or at least deserving of more detail.

I did not review the abstracts; please make sure to adjust them as needed according to modifications resulting from suggestions or comments.

When stating that the outbreak was triggered by the arrival of a ship from Belém, Brazil, I would like to see data on the occurrence of yellow fever in that city at the time (if available). There may be information on this in A história da Febre Amarela no Brasil, authored by Dr. Odair Franco (available at the link above) (I’m not sure that it contains that type of information).

Please review the entire text to correct the use of the abbreviations YF and YFV. On numerous occasions throughout the manuscript, the authors refer to outbreaks/cases of YFV, when the correct term would be outbreaks/cases of YF.

Reviewer #4: I consider this study to be highly original,and the manuscript is very well written.The description of the methods,incluinding the study area, the variable collected, and the application of the inference model is particularyly clear and well presented. In the Discussion section, the authors could be more direct in explaining how the data obtained in the study coul be useful for advancing the understanging of yellow fever.

Reviewer #5: This study reviewed and analysed historical records to recreate the spatial dispersal of YF during the 1857 epidemics in Lisbon Portugal. The study also investigated how environmental and sociodemographic factors of the population to this spread. The authors impressively explained their methodology, provided adequate analysis and highlighted limitations in their methodology and ensuing results.

PLOS authors have the option to publish the peer review history of their article (what does this mean? ). If published, this will include your full peer review and any attached files.

**Do you want your identity to be public for this peer review?** For information about this choice, including consent withdrawal, please see our Privacy Policy .

Reviewer #1: No

Reviewer #2: No

Reviewer #3: No

Reviewer #4: No

Reviewer #5: No

**Figure resubmission:**
---

## [Decision Letter · Decision Letter 1]

19 Feb 2026

Dear Li,

We are pleased to inform you that your manuscript 'Spatial and Social Determinants of the 1857 Yellow Fever Epidemic in Lisbon' has been provisionally accepted for publication in PLOS Neglected Tropical Diseases.

Best regards,

Ran Wang, M.D.

Academic Editor

Álvaro Acosta-Serrano

Section Editor

Shaden Kamhawi

co-Editor-in-Chief

Paul Brindley

co-Editor-in-Chief

Reviewer's Responses to Questions

**Key Review Criteria Required for Acceptance?**

**Methods**

-Are the objectives of the study clearly articulated with a clear testable hypothesis stated?

-Is the study design appropriate to address the stated objectives?

-Is the population clearly described and appropriate for the hypothesis being tested?

-Is the sample size sufficient to ensure adequate power to address the hypothesis being tested?

-Were correct statistical analysis used to support conclusions?

-Are there concerns about ethical or regulatory requirements being met?

Reviewer #3: Yes

Reviewer #4: Yes, the study design, described in the materials and methods section, is appropriate for achieving the objectives and well-developed in relation to the hypotheses.

The study population is clearly described. It is the population affected by yellow fever whose cases were extracted from the 1859 report of the Extraordinary Public Health Council of Portugal.

The statistics used for the samples were adequate to the historical data available for the analysis conducted. And yes, the statistical inference was sufficient to test the hypotheses and support the conclusions.

There are no concerns regarding ethical and regulatory requirements.

**Results**

-Does the analysis presented match the analysis plan?

-Are the results clearly and completely presented?

-Are the figures (Tables, Images) of sufficient quality for clarity?

Reviewer #3: Yes

Reviewer #4: Yes, the analysis presented corresponds to the plan described in the materials and methods section.

The results meet the stated objectives. The figures, which include graphs, diagrams, and maps, are clear and contribute to the understanding of the presented results.

**Conclusions**

-Are the conclusions supported by the data presented?

-Are the limitations of analysis clearly described?

-Do the authors discuss how these data can be helpful to advance our understanding of the topic under study?

-Is public health relevance addressed?

Reviewer #3: Yes

Reviewer #4: This study addresses spatial and social determinants related to yellow fever and its urban vector, which are extremely relevant to public health for any country that has recorded cases of the disease affecting human beings.

The conclusions are supported by the results, and the study's limitations were clearly reported in the discussion.

Yes, the authors discuss how the data obtained may contribute to future research.

**Editorial and Data Presentation Modifications?**

Reviewer #3: I believe the only modification that might still be necessary would be to slightly condense the manuscript, particularly the discussion section. However, I believe this should be a decision for the editors.

Reviewer #4: As a further suggestion, I would add two points: care should be taken not to use terms from the discussion in the description of the results and/or the comparison of the results with recent work. And do not use only the genus to designate the species Ae. aegypti. As I mentioned in the first review, the genus Aedes contains other species.

I add that the article has undergone a substantial improvement in its discussion, and the suggestions I offered were accepted by the authors. I therefore consider the work accepted for publication.

**Summary and General Comments**

Reviewer #3: I must say that I am very pleased with the effort and dedication the authors have shown in evaluating the reviewers’ recommendations and adopting those that were relevant for improving the manuscript. It was undoubtedly a significant effort. I am fully satisfied with how the authors addressed the suggestions, comments, and even the criticisms, and I see no further relevant issues that would lead me to suggest new modifications.

Reviewer #4: I consider this article well-written and original. The description of the methods, including the study area, variables collected, and the application of the inference model, is particularly clear. The results are well described according to their figures, and the discussion is well-developed. This results in acceptance for publication according to the aforementioned item.

---

## [Editor Report · Acceptance letter]

Dear Li,

We are delighted to inform you that your manuscript, "Spatial and Social Determinants of the 1857 Yellow Fever Epidemic in Lisbon," has been formally accepted for publication in PLOS Neglected Tropical Diseases.

Best regards,

Shaden Kamhawi

co-Editor-in-Chief

Paul Brindley

co-Editor-in-Chief
